# Comprehensive Control Strategy of Fuel Consumption and Emissions Incorporating the Catalyst Temperature for PHEVs Based on DRL

**Guangli Zhou [1], Fei Huang [1], Wenbing Liu [1], Chunling Zhao [2], Yangkai Xiang [2] and Hanbing Wei [2,*]**

[1]  China Road and Bridge Engineering Co., Ltd., Beijing 100022, China
[2]  School of Mechatronic &Vehicle Engineering, Jiaotong University, Chongqing 400074, China
[*]  Correspondence: hbwei@cqjtu.edu.cn

**Abstract:** PHEVs (plug-in hybrid electric vehicles) equipped with diesel engines have multiple model transitions in the driving cycle for their particular structure. The high frequency of start–stop of a diesel engine will increase fuel consumption and reduce the catalytic efficiency of SCR (Selective Catalyst Reduction) catalysts, which will increase cold start emissions. For comprehensive optimization of fuel consumption and emissions, an optimal control strategy of PHEVs that originated from the PER-TD3 algorithm based on DRL (deep reinforcement learning) is proposed in this paper. The priority of samples is assigned with greater sampling weight for high learning efficiency. Experimental results are compared with those of the DP (dynamic programming)-based strategy in HIL (hardware in loop) equipment. The engine fuel consumption and $NO_X$ emissions were 2.477 L/100 km and 0.2008 g/km, nearly 94.1% and 90.1% of those of the DP-based control strategy. By contrast, the fuel consumption and NOx of DDPG (Deep Deterministic Policy Gradient)-based and TD3(Twin Delayed Deep Deterministic Policy Gradient) -based control strategy were 2.557, 0.2078, 2.509, and 0.2023, respectively. By comparative results, we can see that the comprehensive control strategy of PHEVs based on the PER-TD3 algorithm we proposed can achieve better performance with comparison to TD3-based and DDPG-based, which is the state-of-the-art strategy in DRL. The HIL-based experimental results prove the effectiveness and real-time potential of the proposed control strategy.

**Keywords:** PHEVs; deep reinforcement learning; control strategy; PER-TD3 algorithm; catalyst temperature

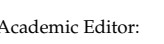



## 1. Introduction

Presently, academia and industry have a consensus that PHEVs (plug-in hybrid electric vehicles) can be a feasible approach to deal with range anxiety and energy savings [1]. To fulfill the requirements of strict emission regulation laws, the SCR (selective catalytic reduction) technique as aftertreatment has been widely applied in diesel engines. Unfortunately, the catalytic efficiency of the SCR of diesel engines is sensitive to the exhaust gas temperature [2]. The temperature of the engine exhaust gas varies abruptly due to the frequent start–stop of diesel engines in PHEVs. Especially for the mode transition from long-range electric driving to hybrid or engine-alone driving, the temperature of SCR is too low to light-off [3]. Consequently, the frequent mode transition of PHEVs may increase the $NO_X$ emissions from the aftertreatment's outlet [4]. Therefore, it is of great significance to develop a comprehensive control strategy to optimize fuel consumption and $NO_X$ emissions simultaneously for PHEVs.

Numerous studies focused on the investigation of the control strategy of PHEVs during the past decade [5]. The rule-based strategy has been widely applied in engineering practice due to its compact structure and high real-time performance [6,7]. However, its performance depends on large number of experiments and expert experience. The optimization-based control strategy implemented the optimization algorithm to minimize the objective function

to deduce the optimal power distribution [8]. The drawbacks were low efficiency and a high computation burden. The learning-based control strategy utilized historical data for model learning and used real-time data for practical applications [9]. Despite adaptation to various conditions, the performance relied strongly on the high accuracy of the vehicle dynamic model and expert experience. Recently, a lot of scholars have applied reinforcement learning to the development of control strategies of HEVs (hybrid electric vehicles). For example, LIU T [10,11] proposed energy management based on Q-learning and DYNA. B Xu proposed an ensemble reinforcement learning strategy to improve the fuel economy of parallel HEVs [12]. C. Xiang developed the adaptive active power filter and attempted to save fuel and prolong the battery life of off-road HEVs by combining the reinforcement learning algorithm [13]. Although their research illuminated the path in the field of the HEV control strategy, the essence of their algorithms still stays in the traditional Q-learning and does not make full use of the function approximation of deep neural networks developed in the field of artificial intelligence in recent years. Specifically, the high-dimension or continuous states of traditional reinforcement learning will lead to dimensional disasters and convergence difficulties. In this context, recent scholars have put forward the state-of-art research of DRL (deep reinforcement learning) and proposed HEV control strategies based on the DQN (deep Q-network) for discrete action space tasks and DDPG (deep deterministic policy gradient) for continuous action space tasks, both of which have demonstrated good policies. DRL shows surpassing performance in the field of those dilemmas as demonstrated by Wang Yong et al. [14]. Moreover, the latest DDPG algorithm that originated from DRL has been proved to effectively reduce fuel consumption [15]. R. Huang proposed a novel EMS (energy management strategy) based on the improved DDPG algorithm with prioritized replay for a power-split plug-in hybrid electric bus to improve the fuel economy as well as the learning efficiency of DDPG [16]. B. Hu investigated an adaptive hierarchical EMS combining heuristic ECMS (equivalent consumption minimization strategy) knowledge and DDPG [17]. Focusing on both the physic and cyber systems, reflecting the dynamic vehicle system in the physical layer, as well as taking full advantage of the outside information in the cyber layer, H. He proposed the cyber-physical system (CPS)-based EMS using DDPG [18]. Meanwhile, the prior valid knowledge trained by HEB was applied to Prius based on deep transfer learning, accelerating the new EMS convergence and ensuring the same initial parameters of the two vehicles' deep neural networks. DDPG can be applied to deal with more complex and continuous action space scenarios. However, the Q value may be overestimated, which may cause the algorithm to fall into suboptimal solutions and lead to non-convergence. For seeking more efficient DRL algorithms, X. Tang proposed a deep Q-network (DQN)-based energy and emission management strategy [19]. Then, two distributed DRL algorithms, namely, asynchronous advantage actor-critic (A3C) and distributed proximal policy optimization (DPPO), were adopted. DQN can effectively solve problems in high-dimension or continuous states. However, on the issue of fuel consumption and emissions for comprehensive optimization of PHEVs, it is necessary to discretize the control actions such as the power of the motor. After the discretization, it is impossible to ensure that the intelligent actor can experience each control action. Meanwhile, its discrete actions cannot be finely adjusted, which will lead to difficulty in convergence. Aiming at improving the efficiency of the automatic development of HEVs' EMSs, R. Lian proposed a transfer learning-based method to achieve the cross-type knowledge transfer between DRL-based EMSs [20]. First of all, massive driving cycles were used to train a DRL-based EMS for Prius. Then, the parameters of its deep neural networks, wherein the common knowledge of energy management is captured, were transferred into EMSs of a power-split bus, a series vehicle, and a series-parallel bus. The motivation of the research is to investigate the versatility for different structures of HEVs other than the training efficiency and optimized performance of DRL.

The issues of the current RL or DRL setting ranging from inappropriate aftertreatment constraint to ineffective and risky exploration make it inapplicable to many industrial control strategy tasks. In this paper, a comprehensive control strategy of economy and

emissions incorporating aftertreatment temperature for PHEVs based on the PER-TD3 algorithm has been proposed. It features the architecture of the AC (action-critic) and the PER (prioritized experience replay) mechanism in order to overcome the difficulty due to the complex action space and continuous state space. Finally, the experimental results are compared with the DP-based control strategy, which has been proved to have global optimal performance [21]. There are two original contributions that clearly distinguish our effort from other studies. (1) First, to the best of our knowledge, this is the first work that investigates how to rationally optimize the fuel consumption and emissions from a diesel engine's aftertreatment in an RL-based control strategy under constraint between the fuel consumption rate and emission performance. (2) Second, unlike most of the standard RL-based procedures where the experience playback mechanism is uniformly and randomly sampled during learning, which will result in repeated sampling and some data not being used as well as ignoring the importance of experience data that will lead to low learning efficiency and over fitting, in this study, the priority of samples is defined to assign more sampling weight to data with a high learning efficiency in order to be applied efficiently. The rest of the article is organized as follows: Section 2 describes the system modelling of the PHEV. Section 3 describes the comprehensive optimal control strategy based on the PER-TD3 algorithm. Section 4 shows the experimental results and discussion. A brief summary is given in Section 5.

## 2. PHEV System Modeling

The configuration of single-shaft parallel PHEVs investigated in the paper is shown in Figure 1. The components and main parameters of the system are shown in Table 1.

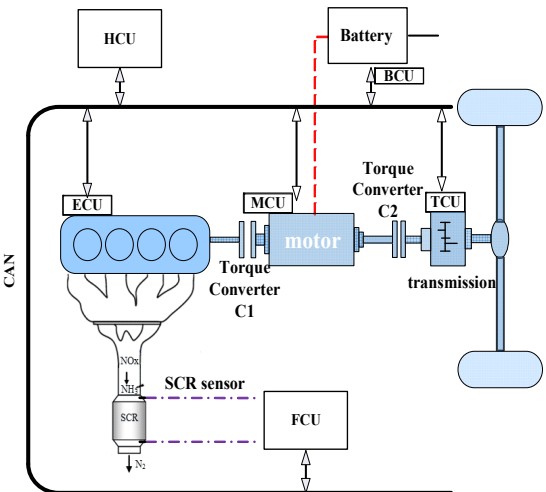

**Figure 1.** Structure of the single-shaft parallel PHEV.

**Table 1.** Main parameters of the PHEV.

| Parameters | Symbols | Values |
|---|---|---|
| Diesel engine | Displacement/L | 1.4 |
| | power/Kw | 72 |
| | Rotation speed r/min | 4200 |
| | Maximum torque/Nm | 184 |
| ISG Motor | Rated power/Kw | 11 |
| | Maximum torque/Nm | 110 |
| Battery | Rated capacity/Ah | 10 |
| | Rated voltage/V | 330 |
| Transmission ratio | $i_1$-$i_5$ | 3.58, 2.02, 1.35, 0.98, 0.81 |
| Main reduction ratio | $i_g$ | 3.947 |

### 2.1. Engine Modelling

In this paper, we apply the experiment-based method to establish the engine model. Only the mapping relationship between the input and output of the engine, such as engine torque, speed with fuel consumption or NO$_X$ emissions is considered in the modelling. Based on the experimental results, the numerical engine fuel consumption and NO$_X$ emission model is established by interpolating the steady-state data as shown in Figures 2 and 3.

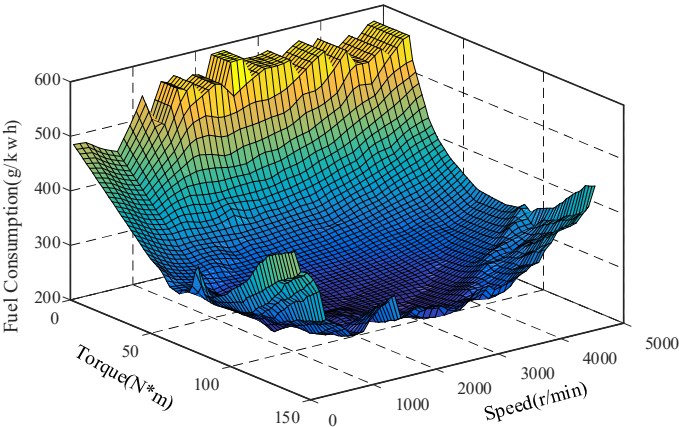

**Figure 2.** Engine fuel consumption map.

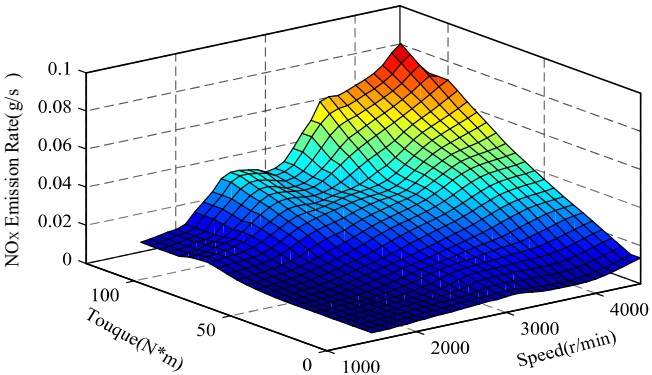

**Figure 3.** Engine NOX emission map.

The numerical expression of the engine modelling can be expressed as

$$\begin{cases} g_e = f_e(\omega_e, T_e) \\ m_{fuel} = \int_0^t f_e(\omega_e, T_e)dt \end{cases} \tag{1}$$

$$\begin{cases} g_{NOx} = f_{NOx}(\omega_e, T_e) \\ m_{NOx} = \int_0^t f_{NOx}(\omega_e, T_e)dt \end{cases} \tag{2}$$

where, $g_e$ is the fuel consumption rate; $m_{fuel}$ is the fuel consumption; $g_{NOx}$ denotes the NO$_x$ emission flow rate; $m_{Nox}$ is the mass of NO$_x$ emission exhaust.

### 2.2. Battery Modelling

Ignoring the temperature's effect on the internal characteristic of the battery, the simplified battery model can be created as shown in Figure 4.

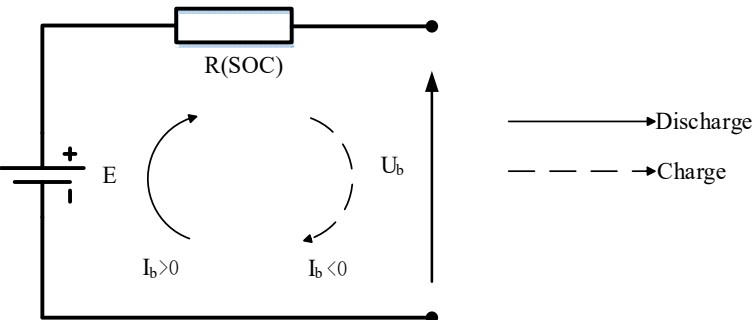

**Figure 4.** Internal resistance model of the battery.

The battery's output voltage is expressed as

$$U_b = V(\text{SOC}) - R(\text{SOC})I_b \tag{3}$$

where $V$ means the open circuit voltage; $R$ means the internal resistance; $I_b$ means the discharge or charge current. The battery's current is expressed as follows.

$$I_b = \frac{V(\text{SOC}) - \sqrt{V^2(\text{SOC}) - 4R(\text{SOC})T_m\omega_m\eta_m}}{2R(\text{SOC})} \tag{4}$$

The battery's SOC is expressed as

$$\text{SOC} = \text{SOC}_0 + \frac{1}{3600}\int_0^t I_b dt \tag{5}$$

### 2.3. SCR's Temperature Modelling

The SCR aftertreatment is one of the efficient techniques to reduce $NO_x$ emissions from diesel engines [22]. The principle of SCR is to catalyze $NO_x$ to $N_2$ and $H_2O$ selectively by catalysts in an oxygen-rich environment. On the premise of the exhaust gas features being incompressible and isentropic, the SCR's temperature can be demonstrated as follows. More details about the SCR dynamic model can be found in [3].

$$\dot{T}_{\text{SCR}} = \frac{M_{\text{exh}}C_{\text{exh}}(T_{\text{eng}} - T_{\text{SCR}}) - h(T_{\text{SCR}}^4 - T_{\text{amb}}^4)}{C_{\text{SCR}}} \tag{6}$$

### 2.4. Vehicle Dynamic Modelling

For PHEVs' control strategy, only the vehicle longitudinal dynamic needs to be investigated. Therefore, the influence of the lateral dynamic can be neglected. Assuming that the vehicle mass is concentrated on the center of gravity, the driving force balance equation can be established as

$$F_t = Mgf\cos\alpha + \frac{C_D A}{21.15}v^2 + Mg\sin\alpha + \delta M\frac{dv}{dt} \tag{7}$$

where $F_t$ means the tractive force; $F_f$ denotes the road friction force from the road; $F_f$ denotes the air drag; $F_i$ denotes the road slope resistance; $F_i$ is the acceleration resistance; $M$ is the mass of the vehicle; $f$ means the coefficient of the road friction; $C_D$ represents the coefficient of air drag; $A$ represents the windward area of the vehicle.

Ignoring the road slope, the required power of the vehicle is calculated as follows according to the vehicle's dynamics.

$$P_{\text{req}} = \frac{1}{3600}\left(Mgfv + \frac{C_D A}{21.15}v^3 + \delta Mv\frac{dv}{dt}\right) \tag{8}$$

### 3. Comprehensive Control Strategy Based on the PER-TD3 Algorithm

*3.1. Fundamentals of RL*

By maximizing the expectation of cumulative reward, the RL seeks the optimal strategy through trial-and-error learning between the agent and the environment [23]. DRL combines the advantage of deep learning (DL) and reinforcement learning (RL) by replacing the Q-value table in RL with the neural network. In this way, the ability of dealing with complex control problems has been improved by a large scale. The basic principle of RL is shown in Figure 5.

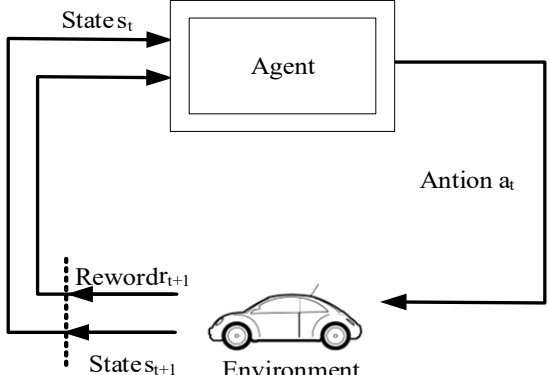

**Figure 5.** Basic principle of RL.

The agent plays a critical role in RL. The agent adopts the strategy at each time step determined by the corresponding action according to the state of the environment $s_t$. Then, the action of the agent interacts with the environment to obtain the corresponding reward $r_{t+1}$ and the next state $s_{t+1}$. The agent improves its behavior strategy according to the size of $r_{t+1}$ so as to maximize the cumulative reward. The complete process of RL is shown in Figure 6.

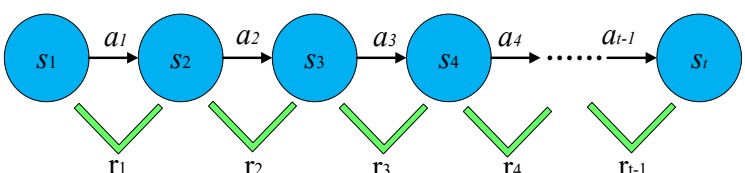

**Figure 6.** The process of RL.

The cumulative reward at time t can be defined as:

$$R(s_t, a_t) + \gamma R(s_{t+1}, a_{t+1}) + \dots \tag{9}$$

The maximum expected reward is defined as:

$$E\left[R_t + \gamma R_{t+1} + \gamma^2 R_{t+2} + \dots\right] \tag{10}$$

The motivation of the agent is to learn an optimal policy $\pi^*$ so that the maximum cumulative reward can be obtained when an action is performed at any time under policy $\pi^*$. The policy is defined as

$$\pi^* = \operatorname{argmax}_\pi Q(s, a), \forall s, a \tag{11}$$

In order to seek the optimal strategy $\pi^*$, the impact of each action on the cumulative reward is determined. The state-action value function is created to measure the state. Then

the action at a certain moment is to estimate the execution of the strategy $\pi^*$ in a certain state. The expected reward is defined as follows:

$$Q(s,a) = E_\pi[R_t + \gamma R_{t+1} + \ldots | s_t = s, a_t = a] \tag{12}$$

The equation above can be abbreviated as follows:

$$Q(s,a) = E_\pi[R_t + \gamma Q(s_{t+1}, a_{t+1}) | s_t = s, a_t = a] \tag{13}$$

Based on the theory above, the features of RL can be demonstrated as follows:

(1) As the state changes, the agent has to perform a different action, and the future state is determined by the current state and the action performed;
(2) The optimization pays more attention to the expected cumulative reward other than the immediate reward;
(3) The agent only needs to know the current state information and then obtain immediate rewards without relying on future inputs and an accurate system's model.

Specifically for the comprehensive control strategy of the PHEV investigated in this paper, it has characteristics as follows:

(1) The control strategy needs to calculate the power of the motor, SCR temperature, battery SOC, and driving mode, etc. The next state is related to the current control action.
(2) The control strategy is to minimize fuel consumption and emissions during the entire driving cycle rather than a certain transient period.
(3) The specific state for the entire trip in advance cannot be achieved in advance. It can only receive the current power demand and vehicle speed. Meanwhile, it can receive the feedback of the current state such as the amount of fuel consumption and emissions.

### 3.2. Modelling of the Control Strategy

The action of the agent is designated as the output power of the motor. The state variables are designed as the required power, SOC, and the temperature the SCR. The objective function is defined as the cumulative return with reward decay:

$$J_t = \sum_{k=0}^{\infty} \gamma^k R(t) \tag{14}$$

The reward function plays an important role in guiding the learning direction of the agent. Particularly for the comprehensive control strategy in this paper, the reward function is demonstrated as follows:

$$R(t) = \omega_1 R_1(t) + \omega_2 R_2(t) + \omega_3(\text{SOC} - 0.4) \tag{15}$$

$$R_1(t) = \begin{cases} -C_{\text{fuel}} & (\dot{m}_{\text{fuel}} \neq 0) and (\text{SOC} \leq 0.8) \\ -(C_{\text{fuel}} + 1) & (\dot{m}_{\text{fuel}} \neq 0) and (\text{SOC} > 0.8) \\ -2 & (\dot{m}_{\text{fuel}} = 0) and (\text{SOC} < 0.3) \\ 0 & (\dot{m}_{\text{fuel}} = 0) and (\text{SOC} \geq 0.3) \end{cases} \tag{16}$$

$$R_2(t) = \begin{cases} -C_{\text{NOx}}(1 - \eta_{\text{NOx}}) & \dot{m}_{\text{fuel}} \neq 0 \\ 0 & \dot{m}_{\text{fuel}} = 0 \end{cases} \tag{17}$$

The control variable is expressed as follows:

$$U(t) = P_{\text{m}}(t) \tag{18}$$

where, $P_{\text{m}}$ is the output power of the tractive motor.

The constraints of the system are expressed as follows according to the system's physical limitation.

$$\begin{cases} \text{SOC}_{\min} \leq \text{SOC}(t) \leq \text{SOC}_{\max} \\ T_{\text{SCR,min}} \leq T_{\text{SCR}}(t) \leq T_{\text{SCR,max}} \\ P_{\text{bat,min}} \leq P_{\text{batt}}(t) \leq P_{\text{bat,max}} \\ P_{\text{m,min}} \leq P_{\text{m}}(t) \leq P_{\text{m,max}} \\ P_{e,\min} \leq P_{\text{e}}(t) \leq P_{e,\max} \\ T_{\text{m,min}} \leq T_{\text{m}}(t) \leq T_{\text{m,max}} \\ T_{e,\min} \leq T_{\text{e}}(t) \leq T_{e,\max} \\ \omega_{\text{m,min}} \leq \omega_{\text{m}}(t) \leq \omega_{\text{m,max}} \\ \omega_{e,\min} \leq \omega_{\text{e}}(t) \leq \omega_{e,\max} \end{cases} \tag{19}$$

The boundary conditions of the system are expressed as follows:

$$\begin{cases} 0.3 \leq \text{SOC}(t) \leq 0.9 \\ 20°\text{C} \leq T_{\text{SCR}}(t) \leq 450°\text{C} \end{cases} \tag{20}$$

where $T_{\text{SCR}}$ means the temperature of the SCR.

Then, the comprehensive optimization problem of PHEVs is transformed into the control action sequence corresponding to the optimal control strategy $\pi^*$ and the optimal state-action value function, which is defined as

$$Q^*(s, a) = \max_\pi E[J_t | s_t = s, a_t = a] \tag{21}$$

### 3.3. Control Strategy Based on PER-TD3

The comprehensive control strategy of the PHEV based on the PER-TD3 algorithm proposed in this paper is shown in Figure 7.

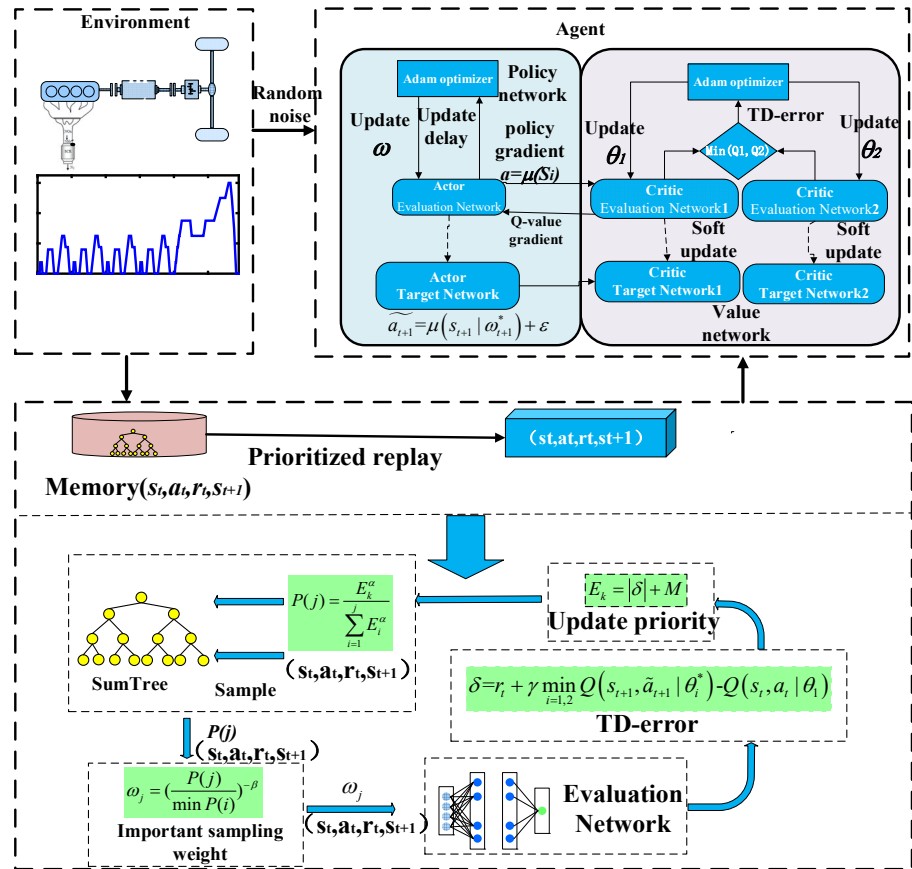

**Figure 7.** Schematic of the control strategy based on PER-TD3.

The control strategy makes use of a deep neural network to fit the policy and action value function, which corresponds to each one network in the figure. They are the actor estimation network $\pi_\omega$, actor target network $\pi_{\omega^*}$, critic estimation network $Q_{\theta_1}$, $Q_{\theta_2}$, critic target network $Q_{\theta_1^*}$, $Q_{\theta_2^*}$. The role and update rules of each network are as follows:

(1) Actor estimation network $\pi_\omega$: updating parameters iteratively, calculating the current optimal action according to the current state $s_t$ when interacting with the environment to generate the next moment state $s_{t+1}$ and the immediate reward;

(2) Actor target network $\pi_{\omega^*}$: calculating the optimal next action $A_{t+1}$ according to the next state $s_{t+1}$;

(3) Critic estimation network $Q_{\theta_1}$, $Q_{\theta_2}$: calculating the action value function $Q(s_t, a_t|\theta_i)$ according to the state and the action output by the actor estimation network, calculating the current $Q$ value gradient and transferring it to the actor estimation network to output the optimal action. At the same time, it is also responsible for the iterative update of the estimated network parameters $\theta_i$, $i = 1, 2$.

(4) Critic target network $Q_{\theta_1^*}$, $Q_{\theta_2^*}$: calculating $Q(s_{t+1}, a_{t+1}|\theta_i^*)$ of the target $Q$ value according to the $s_{t+1}$ and $a_{t+1}$ at the next moment. $\theta_i^*$ is the network parameter of the critical target network.

The updated parameter of the actor target network and two critic target networks adopt the method of soft update. That is, each updated parameter approximates the estimated network parameters with tiny variation. The approximation can be expressed as

$$\begin{aligned} \omega^* &\leftarrow \tau\omega + (1-\tau)\omega^* \\ \theta_i^* &\leftarrow \tau\theta_i + (1-\tau)\theta_i^* \end{aligned} \tag{22}$$

where $\tau$ is the updated coefficient, $\tau \ll 1$. In this paper, it is 0.001.

The critic estimation network iteratively updates the parameters by minimizing the loss function. The loss function is defined as the square of the error between the target $Q$ value and estimated $Q$ value. The optimization algorithm implements the value update of the network parameters by minimizing the loss function. The formula of the loss function is written as follows:

$$y(t) = r_t + \gamma \min_{i=1,2} Q(s_{t+1}, \widetilde{a_{t+1}}|\theta_i^*) \tag{23}$$

$$L(\theta_i) = E\left[(y_t - Q(s_t, a_t|\theta_i))^2\right] \tag{24}$$

Among them, $y(t)$ is the target $Q$ value, $Q(s_{t+1}, a_{t+1}|\theta_i^*)$ is the output of the two critic target networks, where the smaller one is used to calculate the target $Q$ value; $Q(s_t, a_t|\theta_i)$ is the output of the two critic estimation networks using adaptive moment estimation (Adam).

The update of the parameters of the actor estimation network $\pi_\omega$ is implemented on the basis of the $Q$ value gradient provided by the critic estimation network. Its loss gradient is defined as

$$\nabla J(\omega) = \frac{1}{n}\sum_{j=1}^{n}\left[\begin{array}{c} \nabla_a Q(s,a|\theta_i)|_{s=s_t, a=\mu'(s_t)} \\ \nabla_\omega \mu'(s|\omega)|_{s=s_t} \end{array}\right] \tag{25}$$

where $\nabla_a Q(s, a|\theta_i)$ is the gradient of the $Q$ value of the critic estimation network, which means that the action of the actor estimation network should shift to the direction for obtaining a larger $Q$ value; $\nabla_\omega \mu'(s|\omega)$ is the gradient of the actor estimation network, which means that the parameter update of the actor estimation network should be increased.

The loss of the actor estimation network can be considered the feedback of the larger $Q$ value. The loss function of the actor estimation network is defined as

$$J(\omega) = -\frac{1}{n}\sum_{j=1}^{n} Q(s,a|\theta_i)|_{s=s_i, a=\mu'(s_i)} \tag{26}$$

In order to improve the robustness of the algorithm, noise with normal distribution is added to the next action $a_{t+1}$ output by the actor target network. Random noise is imposed on the control action by the actor estimation network in order to ensure that the training process utilized all of the training data, which can be expressed as

$$\widetilde{a_{t+1}} = \mu^*(s_{t+1}) = \mu\left(s_{t+1}\middle|\omega_{t+1}^*\right) + \varepsilon \tag{27}$$

$$\mu'(s_t) = \mu(s_t|\omega_t) + \varepsilon \tag{28}$$

where $\varepsilon$ is the imposed random noise.

The actor strategy network and critic value network are established by using a 5-layer fully connected neural network. The number of neurons of the input layer for the actor policy network and critic value network is 3 and 4, respectively. The number of neurons of the hidden layer for the two networks is 30, 100, and 30. The number of neurons of the output layer is 1 according to the action output of the policy function $\mu(s_t|\omega_t)$ and the action value function $Q(s_t, a_t|\theta_i)$. ReLU is adopted as an activation function.

The brief procedure of PER is to discrete uniform sampling unevenly first and then assign the priority of samples with greater sampling weight for high learning efficiency so that all data can be efficiently used. The PER-TD3 algorithm needs to define the priority of the experience samples during learning. The definition of the experience priority makes use of TD-error, which is defined as the difference between the target $Q$ value and the current estimated $Q$ value, which is expressed as $\delta$ shown in Equation (30). The larger absolute value of TD-error indicates a greater learning improvement space. In other words, the larger the update amount of the agent, the more its samples need to be learned and the higher priority the sample has.

$$y(t) = r_t + \gamma \min_{i=1,2} Q(s_{t+1}, \widetilde{a}_{t+1}|\theta_i^*) \tag{29}$$

$$\delta = y(t) - Q(s_t, a_t|\theta_1) \tag{30}$$

Among them, $y(t)$ is the target $Q$ value, $Q\left(s_{t+1}, \widetilde{a}_{t+1}|\theta_i^*\right)$ is the output of the two critical target networks $Q_{\theta_1^*}$, $Q_{\theta_2^*}$; the smaller one is used to calculate the target $Q$ value; $Q(s_t, a_t|\theta_1)$ is the output of the critical estimation network $Q_{\theta_1}$.

The advantage of the PER-TD3 algorithmic is to playback the samples with high priority. In this way, the algorithm can accelerate the convergence speed. However, some samples with higher priority may be played back repeatedly. As result, the special experience samples that are 0 cannot be sampled and played back, which will lead to a single sample [24]. Insufficient diversity of samples will lead to the overfitting of the algorithm. To avoid this condition, a normal number $\varsigma$ is introduced to make up for the diversity. The sample priority can be expressed as

$$E_k = |\delta| + \varsigma \tag{31}$$

where $E_k$ is the playback priority of the experience sample; $M$ is a small positive number, which ensures that the experience sample, which is 0, can be sampled and played back.

In order to deal with the problem that $|\delta|$ fluctuates greatly when the environment is random, the sampling probability is defined as Equation (32).

$$P(j) = \frac{E_k^\alpha}{\sum\limits_{i=1}^{j} E_i^\alpha} \tag{32}$$

where $P(j)$ is the sampling probability of the sample; $\alpha$ is the algorithm hyper parameter, which determines the randomness of the sampling. When $\alpha$ is 0, it means uniform sampling. When $\alpha$ is 1, it means the sampling depends on priority completely.

Figure 8 shows a concise comparison of uniform sampling and playback based on prior experience. Playback with priority experience will change the probability distribution of experience segments and introduce bias. Therefore, the loss function should be modified with important sampling weights to make sure that samples have the same influence on the gradient descent.

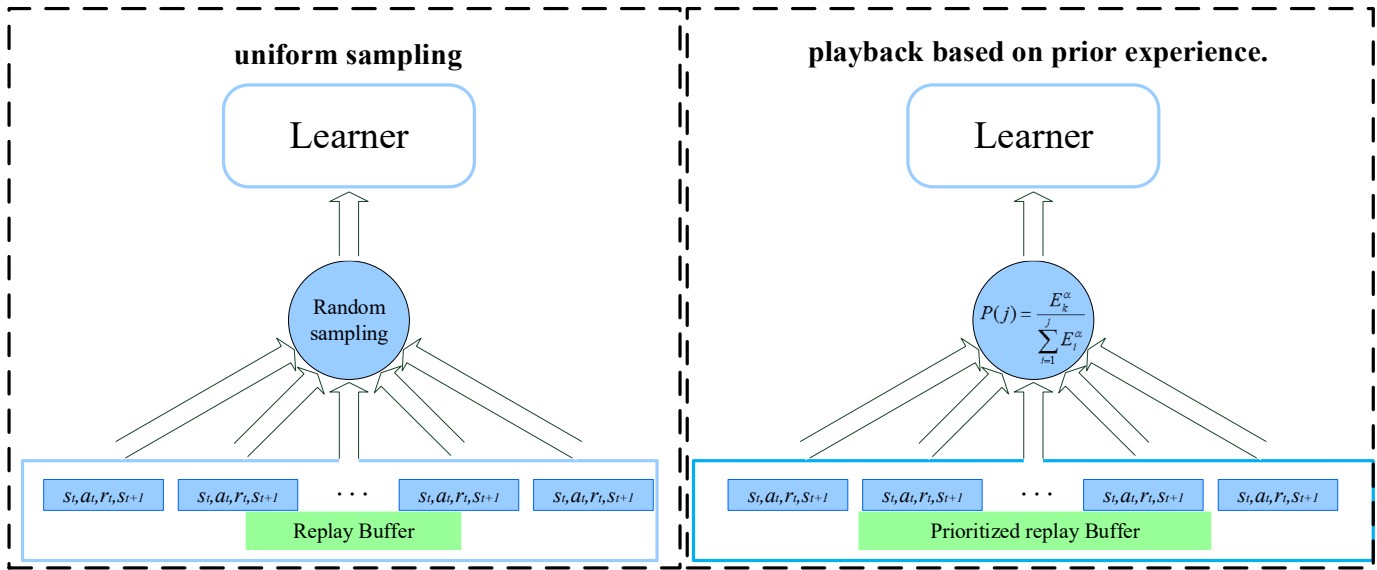

**Figure 8.** Comparison of the two sampling methods.

The sampling weights is shown in Equation (33), where $\beta$ is a hyper parameter used to smoothen the high variance weights to offset the degree to which prior experience replay affects the results. The optimization of the loss function of its critic estimation network is shown in Equation (34). The flowchart of the algorithm is shown in Algorithm 1.

$$\omega_j = \left(\frac{P(j)}{\min P(i)}\right)^{-\beta} \tag{33}$$

$$L(\theta_i) = E\left[\omega_j\left(r_t + \gamma \min_{i=1,2} Q(s_{t+1}, \widetilde{a}_{t+1}|\theta_i^*) - Q(s_t, a_t|\theta_i)\right)^2\right] \tag{34}$$

where $\omega_j$ is the sampling weight; $P(j)$ is the sampling probability.

---

**Algorithm 1. Calculation Process of the Strategy Based on PER-TD3 Algorithm**

---

1: initialization:

actor estimation network $\pi_\omega$, critic estimation network $Q_{\theta_1}$, $Q_{\theta_2}$: $\omega$, $\theta_1$, $\theta_2$

parameter of $\pi_{\omega^*}$, $Q_{\theta_1^*}$, $Q_{\theta_2^*}$: $\omega^* \leftarrow \omega$, $\theta_1^* \leftarrow \theta_1$, $\theta_2^* \leftarrow \theta_2$

prioritize playback of the default data structure of the experience pool SumTree

2: For episode = 1:M do

3:     get initial state: $P_{req}$, SOC, $T_{SCR}$

4:     For $t$ = 1:T do

5:         Pick $a_t = \mu(s_t|\omega_t) + \varepsilon\prime$ based on the current strategy and exploration noise

6:         Execute at to get reward rt, and observe the state of the system at the next moment

7:         Store $e_t = (s_t, a_t, r_t, s_{t+1})$ in the priority replay experience pool SumTree

8:          if experience pool data is greater than 1000

9:         n samples $(s_t, a_t, r_t, s_{t+1})$ are drawn from SumTree; the probability of each sample is based

on $P(j) = \dfrac{E_k^\alpha}{\sum\limits_{i=1}^{j} E_i^\alpha}$, and the weight $\omega_j = (\dfrac{P(j)}{\min P(i)})^{-\beta}$ is calculated simultaneously.

10:         Calculate the target action: $\tilde{a}_{t+1} = \mu(s_{t+1}|\omega_{t+1}^*) + clip(N(0,\sigma), -c, c)$

11:         Calculate the target $Q$ value:

$$y(t) = r_t + \gamma \min_{i=1,2} Q(s_{t+1}, \tilde{a}_{t+1}\Big|\theta_i^*)$$

12:         Calculate the loss for $Q_{\theta_1}$, $Q_{\theta_2}$: $L(\theta_i) = E\Big[\omega_j(y_t - Q(s_t, a_t|\theta_i))^2\Big]$, Update parameter $\theta_i$

with the Adam optimization algorithm

13:         Recalculate TD-error for all samples, $\delta = y(t) - Q(s_t, a_t|\theta_1)$; Update priority

$E_k = |\delta| + M$ of all nodes in SumTree

14:          if t mod d then

15:             Calculate the loss of $\pi_\omega$: $J(\omega) = -\dfrac{1}{n}\sum\limits_{j=1}^{n} Q(s, a|\theta_i)|_{s=s_i, a=\mu\prime(s_i)}$; Update parameter $\omega$

with Adam optimization algorithm

16:             Update the parameters of the target network $\pi_{\omega^*} Q_{\theta_i^*}$:

$\omega^* \leftarrow \tau\omega + (1-\tau)\omega^*$

$\theta_i^* \leftarrow \tau\theta_i + (1-\tau)\theta_i^*$

17:             End if

18:         End if

19:         enter the next state

20:     End for

21: End for

---

## 4. Experimental Results and Discussion

In order to validate the comprehensive control strategy of fuel consumption and emissions we proposed in this paper, a Hardware-in-Loop (HIL) experimental platform was constructed by using the Matlab/Simulink module in combination with the RTI module provided by dSPACE, as shown in Figure 9.

In this experiment, a low-cost MC9S12XS128 micro control unit (MCU) is employed as the master chip of the vehicle control unit (VCU). The control strategy developed in Matlab/Simulink can be compiled into C++ code by Targetlink software provided by the dSPACE company. Both the strategy and driving program are embedded in the VCU through Codewarrier compilation software. As a virtual controlled target, the vehicle dynamic model established by Matlab/Simulink is implemented in the dSPACE Autobox real-time system. During the process of the IL test, the A/D module of Autobox is used to collect the signals of the accelerator pedal and the brake pedal produced by the dynamic model in dSPACE Autobox and inject them into the VCU. The CAN receiver module in dSPACE Autobox is applied to receive the optimal power distribution, the working mode of the vehicle, the transmission gear, and other signals produced by the VCU. Similarly, the CAN writing module is used to send the relevant signals of the battery and aftertreatment to the VCU, including the battery SOC, battery temperature, voltage, and SCR temperature produced by the vehicle dynamic model. In this way, the interactive communication between the VCU and virtual vehicle target can be implemented. The function of the Con-

troldesk software is that the historic experimental data can be recorded and the variable of the control strategy can be adjusted conveniently. The HIL-based experiment is conducted under the same operating conditions as offline simulation. The training evaluation and analysis of the PER-TD3-based strategy are carried out under the NEDC driving cycle. The relevant parameters are shown in Table 2.

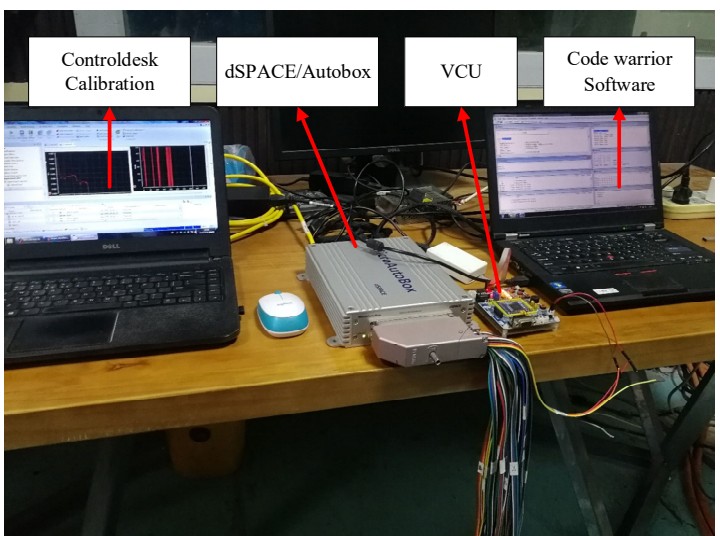

**Figure 9.** Hardware-in-loop test platform.

**Table 2.** Parameters of the PER-TD3-based strategy.

| Parameters | Value |
|:---:|:---:|
| Minibatch n | 64 |
| Discount factor $\gamma$ | 0.9 |
| Update factor $\tau$ | 0.001 |
| Replay buffer capacity | 118000 |
| Actor network learning rate | 0.001 |
| Update delay d | 3 |
| Critic network learning rate | 0.002 |

Figure 10 shows the cumulative rewards of the DDPG-based, TD3-based, and PER-TD3-based strategy during the training period. By comparison, it can be observed that in the early training stage, the curve of the cumulative reward value of the PER-TD3-based strategy shows a more obvious upward trend. Considering that the ideal criterion of PER-TD3 is that the larger the TD-error, the greater the weights of the sample; thereafter, the more the updated amount the agent has, the higher the learning efficiency and the faster the convergence. This indicates that the agent of the PER-TD3-based strategy can optimize the network parameters continuously as well as adjust the strategy and then gradually deduce the optimal strategy. The essential cause is that the priority experience playback not only considers the priority of each sample but also optimizes the loss function to eliminate the bias brought about by non-uniform sampling. In the later training stage, the PER-TD3-based strategy converges gradually and both the reward and reward value gradually stay at the peak value. The DDPG-based and TD3-based strategy gradually converge at 1600 and 1200 rounds, respectively. Comparatively, the PER-TD3 -based strategy begins to gradually converge at around 1000 rounds and achieves a larger reward value. Consequently, the convergence speed and effect of learning optimization for the PER-TD3-based strategy are significantly improved compared to that of the TD3-based and DDPG-based strategies.

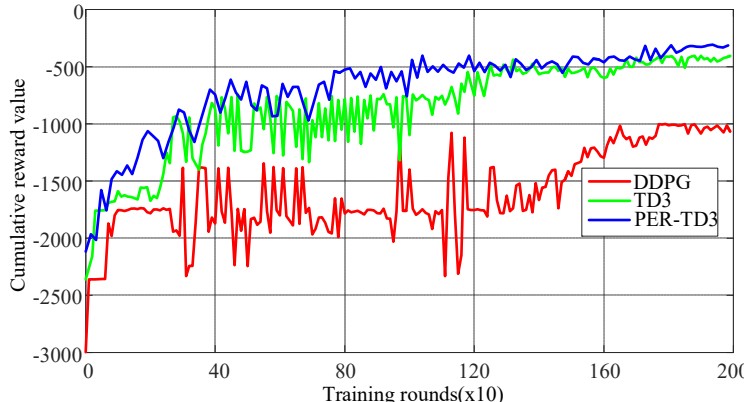

**Figure 10.** Cumulative return value.

Figures 11 and 12 show the comparative curves of the battery's SOC and motor power of the PER-TD3-based and DP-based control strategies. It can be seen that compared with the TD3-based strategy; the SOC of the PER-TD3-based strategy press closer to that of the DP-based control strategy. Meanwhile, the motor power distribution of the PER-TD3-based control strategy approximately coincides with that of the DP-based strategy. It can be seen that the proposed strategy features a more likely global optimal effect than the TD3-based strategy.

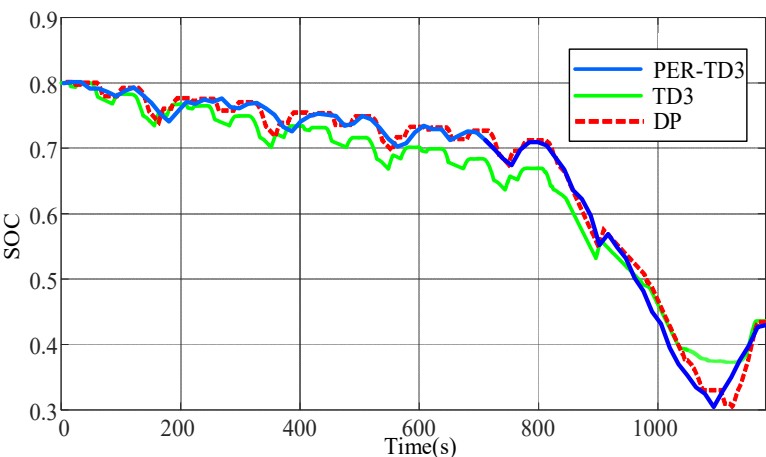

**Figure 11.** SOC curve of TD3 and DP.

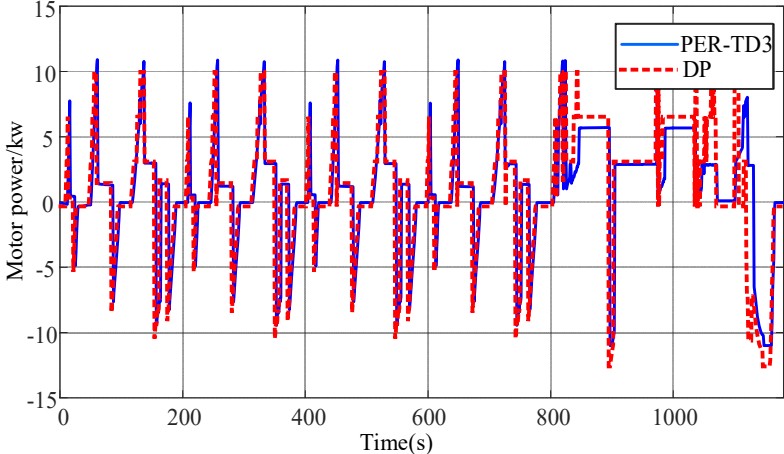

**Figure 12.** Motor power distribution of PER-TD3 and DP.

Figures 13 and 14 show the temperature and catalytic conversion efficiency of the SCR catalyst. It can be seen that, in comparison to the TD3-based strategy, the temperature and conversion efficiency of the SCR catalyst of the PER-TD3-based strategy generously overlap with those of the DP-based strategy, approaching those of the DP-based strategy more closely than those of the TD3-based strategy. In particular, the SCR catalytic efficiency of the PER-TD3-based strategy arrives at the peak value faster than that of the TD3-based strategy during the cold-start period (the first 300 s as shown in the figure). Therefore, the catalytic efficiency can be significantly improved.

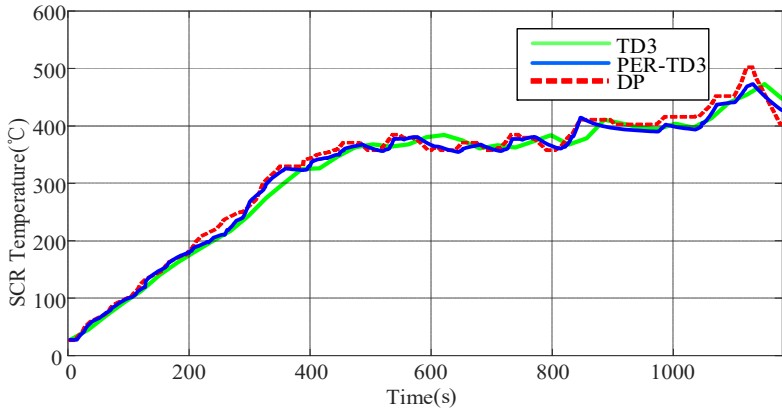

**Figure 13.** SCR temperature of different strategies.

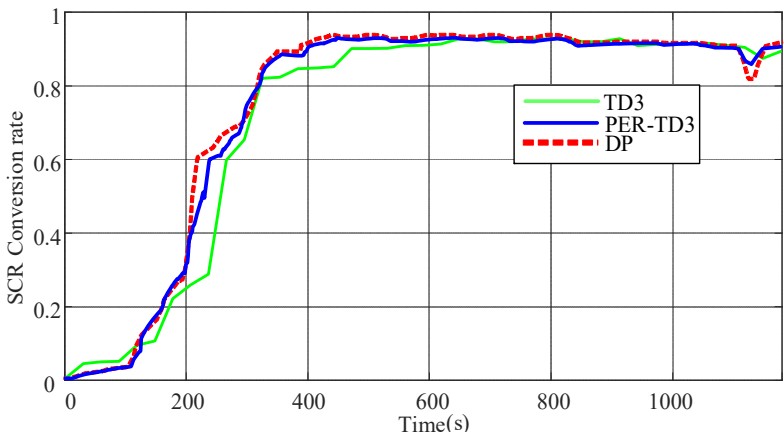

**Figure 14.** SCR catalytic efficiency of different strategies.

Figures 15 and 16 illustrate the distribution of the engine workload on the fuel consumption and NO$_X$ emissions map. It can be seen that the engine workload is nearly constrained in the medium load region. In other words, the average operation efficiency of the engine can be improved under this condition. At the same time, the workload distribution of the PER-TD3-based control strategy is close to that of the DP-based strategy, which realizes the optimal policy behavior. The results of the PER-TD3-based, TD3-based, DDPG-based, and DP-based control strategies are listed in Table 3. The engine fuel consumption and NO$_x$ emissions are 2.477 L/100 km and 0.2008 g/km, nearly 94.1% and 90.1% of those of the DP-based control strategy. By contrast, the fuel consumption and NO$_x$ of the DDPG-based and the TD3-based control strategy are 2.557, 0.2078, 2.509, and 0.2023, respectively. It can be seen that the comprehensive control strategy of the PHEV based on the PER-TD3 algorithm we proposed can achieve the best performance with comparison to TD3 and DDPG, which is the state-of-the-art strategy in DRL.

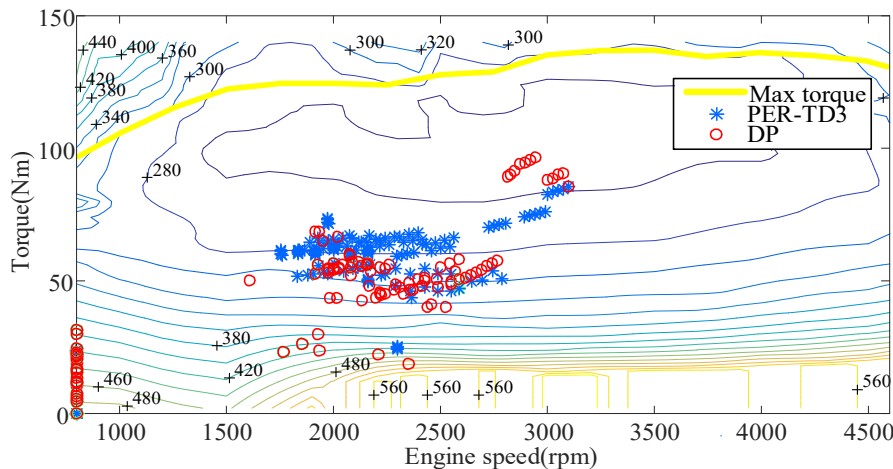

**Figure 15.** Distribution of the engine workload on the fuel consumption map.

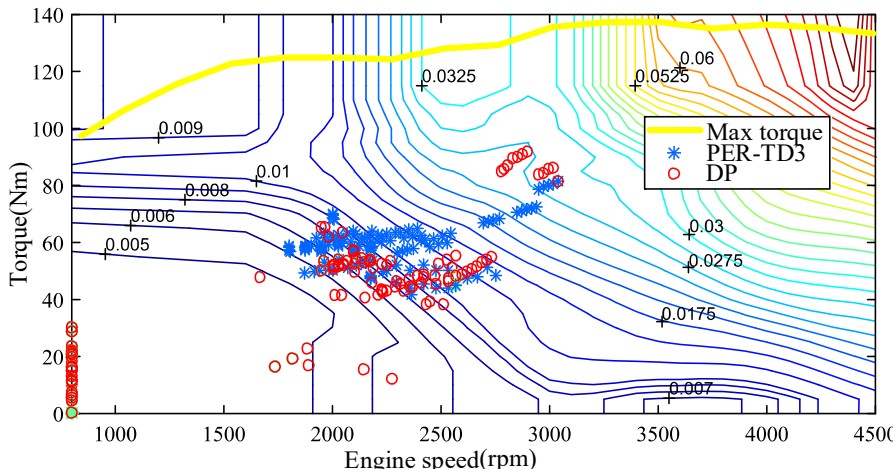

**Figure 16.** Distribution of the engine workload on the NO$_x$ emissions map.

**Table 3.** Comparison of the performance for four strategies.

| Strategy | Fuel Consumption L/100 km | NO$_x$ g/km |
|----------|---------------------------|-------------|
| DP | 2.331 | 0.181 |
| DDPG | 2.557 | 0.2078 |
| TD3 | 2.509 | 0.2023 |
| PER-TD3 | 2.477 | 0.2008 |

## 5. Conclusions

In order to obtain the comprehensive performance of fuel consumption and emissions from diesel engines after treatment of PHEVs, a comprehensive control strategy based on the PER-TD3 algorithm is proposed in this paper. For high learning efficiency, the priority of samples is assigned with greater sampling weight. Offline training was conducted under the NEDC driving cycle to achieve the optimal motor power distribution. The HIL-based experimental results show that the control strategy proposed in this paper obtained excellent fuel saving and emission reduction effects. The fuel consumption was 2.477 L/100 km, which was 94.1% of that of the DP-based strategy. The NO$_X$ emissions of the outlet of the SCR catalyst were 0.2008 g/km, i.e., 90.1% of those of the DP-based strategy. Meanwhile, it achieved the best performance with comparison to TD3 and DDPG, which is the state-of-the-art strategy in DRL. The experimental results demonstrate the effectiveness

and potential capability of the PER-TD3-based strategy we proposed in application of PHEVs' energy management.

**Author Contributions:** Conceptualization, H.W.; methodology, G.Z. and C.Z.; software, F.H.; validation, H.W. and G.Z.; formal analysis, C.Z.; data curation, W.L.; writing—original draft preparation, F.H., G.Z. and W.L.; writing—review and editing, H.W.; visualization, G.Z.; supervision, H.W.; project administration, C.Z.; funding acquisition, Y.X. All authors have read and agreed to the published version of the manuscript.

**Funding:** This research was funded by the National Natural Science Foundation of China, grant number 52172381, and Chongqing Science and Technology Innovation and Application Development Project of China, grant number cstc2019jscx-zdztzx0014.

**Acknowledgments:** The authors would also like to thank the reviewers for their corrections and helpful suggestions.

**Conflicts of Interest:** The authors declare no conflict of interest.

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
