# Peer review of "Comprehensive Control Strategy of Fuel Consumption and Emissions Incorporating the Catalyst Temperature for PHEVs Based on DRL"

_energies, doi:10.3390/en15207523_

Round 1

Reviewer 1 Report

In this article, the authors present a control strategy involving the catalyst temperature of PHEVs to improve fuel consumption and NOx emissions. They validate the presented algorithms using measurement results from a HIL test bench.

Comments:

In the title, you write about a control strategy, but you don't say for what. The title needs to be improved.

In principle, equations should be centered in the text, for some equations the format needs to be improved.

Urgently improve the writing of mathematical variables in the text.

Line 88: Temperature dependent battery performance is significant in all EV applications. Briefly explain why in your model the temperature dependence of the battery can be ignored.

Pay attention to the order of explanation of abbreviations.

Pay attention to spaces, e.g. Fig5 -> Fig. 5

The strategy in equation 17 cannot be understood. Explain the case distinction as a function of SOC.

Explain the boundary conditions for the SOC in equation 21.

Line 218: Why tau=0.001 in your paper?

The diagrams in Figs. 7 and 8 are hard to read.

Table 2 does not show a flowchart; the content cannot be understood.

Section 4. From my point of view, this is not a HIL test, where is the hardware? I would refer to the test setup as SIL. Maybe I am missing something. If so, the experimental setup needs to be described in more detail.

Fig. 11 – 14: How can I see the improvement of the presented method?

The experimental results are described very concisely. A more detailed explanation would be desirable. In particular, the versatility of the presented method should be emphasized more.

The conclusion is very short; the outlook on further work is missing.

Subsection 3.3 appears very mathematical and isolated from the actual problem. The authors should try to present the novel approach in its entirety with respect to the problem in a way that is comprehensible to the reader.

Author Response

thanks for your kindly susgestion. we have update the manuscript and upload the response file for each reviewer's comments.Please see the attachment.

Reviewer 2 Report

The manuscript has been written in a good manner, however, only a few comments need to be considered.

For the introduction section;

In the Introduction section, it is necessary to substantiate the relevance of this scientific topic to a practical problem in the present. Arguments need to be made that it is very important to conduct research on this topic and that the results of such research are necessary for practice.

In the Results section, figures 7 and 9 need to be clearly presented 

Author Response

thanks for your kindly suggestion. We have updated the manuscript and uploaded the response file for each reviewer's comments.Please see the attachment.

Reviewer 3 Report

The subject of the paper is good, but presentation is poor and results insufficient developed.

Some of the terms used into the text are not explained or explanations are not correlated exactly with the text where are used.

Must be given a higher importance to the experimental results and their discussion.

The conclusions must be improved to underline the relevance of the study for researchers interested in.

The English language need fine spelling corrections.

Author Response

(The authors gave the same response as above.)

Reviewer 4 Report

This paper introduces an optimal control strategy of PHEV originated from PER-TD3 algorithm which is on the basis of deep reinforcement learning. After going through the paper, I found some concerns, as listed below, which must be considered:

1.      In the title, I prefer to use full names instead of abbreviations.

2.      In the abstract, focus on the advantages of the proposed method with respect to the obtained results.

3.      Most of the ideas written were already described in many literatures. The Authors tried to compile it but lack of the enhancement of the interrelation analysis between the references. It is advised that the authors give a deeper analysis on how these ideas become more applicative strategies so that they can contribute to the next step of implementation.

4.      In the introduction, you should define the abbreviations in advance, even if you did that in the abstract.

5.      Major clarifications and explanations are needed make the contributions of the paper clearly stand out. There exist many works that are focusing on energy management considering different distributed means of generation, responsive loads, EV, storage units, and in the presence of uncertainties. The novelty of the present work should be well stated and justified. The new author's contribution should be justified regarding the previous works in the literature (such as https://doi.org/10.1016/j.ijepes.2021.106845, https://doi.org/10.3390/en14030569 and many more). The literature review should be updated to help readers better understand the subject matter and novelty aspects of this work compared to the recently published works.

6.      Have you considered the velocity constraint in this research; and what about other constraints?

7.      There are several up-to-date approaches for the idea. Authors should look for these approaches, compare the results and prove their idea. This is the major concern.

8.      Author must justify the choice of solution method, i.e. DRL. What is the rationale behind using such optimization method and not to use other mathematical-based approaches? How do authors guarantee optimality of the obtained solutions?

9.      There are several grammatical mistakes. Please work close to a native English speaker to refine the language of this paper.

Author Response

(The authors gave the same response as above.)

Round 2

Reviewer 1 Report

The authors have partially addressed the suggestions for improvement.

Reviewer 4 Report

The paper has been improved